# SleepMG: Multimodal Generalizable Sleep Staging with Inter-modal Balance of Classification and Domain Discrimination

## ABSTRACT

Sleep staging is crucial for sleep tracking and health assessment. Polysomnography (PSG), containing multiple modalities such as electroencephalography, electrooculography, electromyography, and electrocardiography, is the fundamental means of sleep staging. However, due to performance differences in both classification and domain discrimination across modalities in PSG, existing domain generalization methods face a dilemma of modal imbalance. To balance inter-modal differences and achieve highly accurate cross-domain sleep staging, we propose **SleepMG**, a **M**ultimodal **G**eneralizable **Sleep** staging method. SleepMG assesses the classification and domain discrimination performances of each modality and further defines the modal performance metrics by calculating the variance between the performance score and the average performance of each modality. Guided by these metrics, the gradients of the classifier and domain discriminator are adaptively adjusted, placing greater emphasis on poorly-balanced modalities while reducing emphasis on well-balanced modalities. Experimental results on public sleep staging datasets demonstrate that SleepMG outperforms state-of-the-art sleep staging methods, effectively balancing multiple modalities as evidenced by the visual experiment of modal imbalance degree. Our code will be released after formal publication.

## CCS CONCEPTS

• **Information systems** → *Multimedia information systems*; • **Computing methodologies** → *Transfer learning*; • **Human-centered computing** → *HCI design and evaluation methods*.

## KEYWORDS

Inter-modal balance, Polysomnography, Sleep staging, Domain generalization, Domain discrimination

## 1 INTRODUCTION

Sleep staging [54] is critical for health assessment and intervention, providing critical information on sleep quality of subjects and assisting in screening for brain and neurological health [5, 62, 12, 45]. Polysomnography (PSG) is multimodal physiological signals collected synchronously from different positions of the subject during sleep. PSG-based sleep staging [51, 25] can effectively monitor sleep progress and assist in diagnosing diseases such as Parkinson's

*ACM MM, 2024, Melbourne, Australia*
© 2024 Copyright held by the owner/author(s). Publication rights licensed to ACM.
ACM ISBN 978-x-xxxx-xxxx-x/YY/MM
https://doi.org/XXXXXXX.XXXXXXX

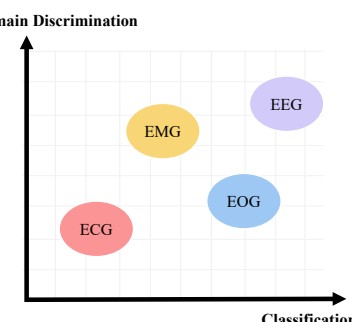

**Figure 1: Varied classification and domain discrimination performances across different modalities of PSG.**

and epilepsy. To ensure consistency, researchers mostly followed the American Academy of Sleep Medicine (AASM) [4] standard, classifying sleep into five stages for conducting studies. Initially, researchers manually staged sleep from the perspective of signal processing and medical knowledge based on the AASM standard. With the rise of deep learning, researchers began to employ automatic feature learning [51, 13, 46] to solve the time-consuming and labor-intensive problem in sleep staging. In sleep research, Convolutional Neural Networks (CNNs) were employed to automatically capture the spatial patterns in different sleep stages [48, 15] and distinctive features in different frequency bands [53], Recurrent Neural Networks (RNNs) [7, 49] and Long Short-Term Memory Networks (LSTMs) [63, 40] were employed to learn the dynamic patterns of signals [7, 49, 63] and capture the discriminative features at different time points [40]. Graph Convolutional Networks (GCNs) [20] were employed to model the association patterns between EEG channels. Sequence-to-sequence models [36, 38] were employed to capture the temporal dependencies between sequences and then learn distinguishing features in sleep staging tasks. Furthermore, the attention mechanism [42, 65, 10] was introduced to capture the differentiating features based on the relative importance of different sleep epochs.

However, as depicted in Figure 1, modal differences exist when multiple modalities of PSG jointly represent the sleep state. To explore the differences, modality-by-modality feature learning and fusion methods [17, 19, 60] have been designed to capture the differentiated features across modalities. Moreover, researchers have developed attention-based fusion techniques [60, 8] to enable multimodal models to focus on more important modalities. However, the attention mechanism amplifies the dominance of stronger modalities, exacerbating the inherent imbalance [35, 11] among them. This imbalance further hinders the model from fully utilizing the potential of all modalities. To balance modalities and make each modality as optimal as possible, Peng et al. [35] balanced two modality-specific feature extractors according to the classification performance of each modality. However, it is regrettable that they

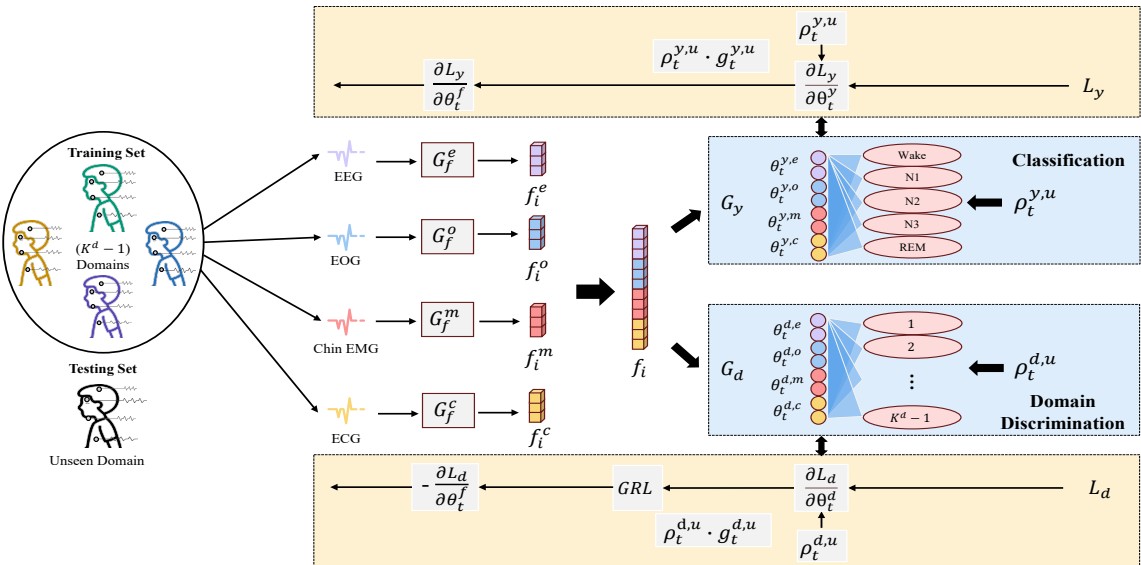

**Figure 2: The pipeline of SleepMG. First, we perform modality-specific feature extraction, followed by the computation of modal performance metrics in classification and domain discrimination. These metrics are then utilized to adaptively influence the gradient backpropagation of the globally shared classifier and domain discriminator. SleepMG quantifies and balances inter-modal differences in two aspects, improving multimodal generalizable sleep staging.**

compared the relative performance of two modalities, making it suitable only for balancing between two modalities. Additionally, balancing modality-specific feature extractors may interfere with the model to decouple and learn modality-related information, a more pronounced phenomenon in multimodal balancing scenarios.

Furthermore, PSG signals exhibit severe subject-dependency and existing generalizable sleep staging methods aim to enhance the cross-subject generalization capability of the model. The pre-training and fine-tuning method [37, 50] in transfer learning was introduced to do the subject personalized calibration. Unfortunately, the method requires significant data and some labeled data from the target subjects, posing inconvenience. As a result, the domain generalization method [57] was integrated. For example, domain adversarial methods [18, 29] were introduced to confuse the discrimination of the domain (i.e., the subject) by backpropagating the negative gradient of the domain discrimination loss, thereby allowing the model to learn domain-invariant information that is beneficial to unseen-domain generalization. Additionally, explicit feature alignment methods [39, 50] were introduced to learn shared feature representations across domains, improving the performance of the model on new fields or unseen data. However, as depicted in Figure 1, modal differences are not only reflected in classification performance, [6, 27], but also in domain discrimination performance [33, 24, 59]. The classification performance varies across different modalities [41], and it differs even more within different subject domains. The above methods enhance the cross-domain generalization of the model. However, none consider the inter-modal difference of domain generalization capability [33, 59], which is also crucial for sleep staging.

To tackle these challenges, we introduce SleepMG, a novel multimodal generalizable sleep staging method through classification

and domain discrimination balancing. SleepMG first quantifies and balances the modal differences in classification and domain discrimination. SleepMG integrates domain adversarial learning into the multimodal feature learning framework and explicitly distinguishes the classification and domain discrimination performances of each modality through a globally shared classifier and discriminator separately. Guided by modal performance metrics, the gradients of model of the classifier and domain discriminator are adaptively adjusted, balancing the emphasis of the model on well and poorly-balanced modalities. SleepMG achieves modality-balanced and improves the multimodal generalizable sleep staging.

To summarize, our contributions are as follows:

- We construct a multimodal generalizable sleep staging model and first introduce metrics to assess the classification and domain discrimination performances of multiple modalities.
- By leveraging modal performance metrics, the gradients of the classifier and domain discriminator are adjusted adaptively, increasing greater emphasis on poorly-balanced modalities while reducing emphasis on well-balanced ones. This adaptive adjustment achieves balance across multiple modalities in two critical aspects.
- Extensive experiments on public datasets show that our proposed SleepMG achieves state-of-the-art sleep staging results, demonstrating its effectiveness in achieving multimodal balance and cross-domain generalization.

## 2 RELATED WORK

This section reviews the related work from two perspectives: sleep staging and generalizable sleep staging.

## 2.1 Automatic Sleep Staging

Sleep staging is crucial in assessing sleep quality and identifying potential neurological disorders [45]. Various methods have been developed to classify sleep stages based on PSG. Early studies relied on manual sleep staging, where experts manually classified sleep stages based on signal processing techniques and medical knowledge, following the AASM standard [3]. The AASM standard categorizes sleep into three main stages: the Wake stage, the Non-Rapid Eye Movement (NREM) stage, and the Rapid Eye Movement (REM) stage. The NREM stage is further divided into three sub-stages: N1, N2, and N3. These stages correspond to different degrees of sleep, resulting in five distinct categories. This manual approach is time-consuming and labor-intensive, prompting researchers to turn to deep learning techniques for automated sleep staging. Researchers utilized CNN [48, 15] to capture local spatial information in PSG data. Additionally, researchers have utilized RNNs [7, 63] and LSTM [48, 44] models to capture the sequential dependencies and temporal dynamics in PSG data. To model the correlations between different channels in PSG data, Jia et al. [20] introduced GCN-based GraphSleepNet, which represents PSG data as a graph to capture complex relationships. To handle the multi-level nature of PSG data, Phan et al. [36, 38] proposed sequence-to-sequence models to capture temporal dependencies across sleep epochs.

PSG includes multiple modalities collected from subjects, such as electroencephalography (EEG) [9, 14, 28, 22], electrooculography (EOG) [21], electromyography (EMG) [58], and electrocardiography (ECG) [32]. To explore the difference and consistency across modalities, researchers have designed modality-specific feature learning [52] modules and corresponding fusion modules [61]. Xiang et al. [58] designed a spatial encoder to capture the high-level shared semantic information between EEG and EMG. Considering the differences of importance across modalities, attention mechanisms have been incorporated into sleep staging models to help the model focus on more important modalities or parts. Zheng et al. [60] also used the attention mechanism to help learn the interdependencies across modalities. However, there are differences across modalities, and attention-based fusion can exacerbate the inter-modal imbalance, resulting in insufficient learning of each modal and limiting the classification capability of multimodal models.

## 2.2 Generalizable Sleep Staging

PSG, as a physiological signal, is highly subject-dependent [34], and the data distribution across subjects varies significantly. In this paper, we explore methods to enhance the out-of-distribution generalization of sleep staging models, aiming to construct a general automated sleep staging model that can be applied across subjects (i.e., domains). We refer to this model as a generalizable sleep staging model [30, 26, 43]. To achieve this, researchers have introduced transfer learning methods. A common approach is the pretraining-finetuning method [26, 56]. Wang et al. [56] trained on a large sleep dataset, MASS, to learn general sleep-related representations, and then fine-tuned on a smaller sleep dataset, sleep-edf sub-dataset. While this method has demonstrated effectiveness, it is not without its drawbacks. The pretraining phase requires a large dataset and substantial computational resources. The finetuning

phase requires a small amount of labeled data, impractical in cross-domain application scenarios such as healthcare. Banluesombatkul et al. [1] used meta-learning for pretraining, which improved data dependency and task generalization to some extent. However, the method increased the computational cost during the pretraining phase. The finetuning phase still requires a small amount of labeled data, making cross-domain generalization unachievable. Tang et al. [50] addressed domain adaptation by explicitly aligning the feature distributions between the source and target domains, removing the reliance on labels in the target domain. However, it still requires data from the target domain and does not solve the cross-domain generalization problem.

To overcome these limitations, researchers integrated the domain generalization method. For example, some researchers incorporated domain adversarial learning [18, 29] into their model. The domain discriminator in domain adversarial played a crucial role in enhancing the cross-domain generalization performance of the model by intentionally confusing the capability of the model to identify different domains. The domain adversarial method has achieved considerable generalization performance. However, there is no specific domain adversarial technique for multimodal sleep staging. Moreover, differences exist across modalities, not only in terms of classification capability but also in domain generalization capability. Quantifying and balancing these two types of inter-modal differences is necessary.

## 3 PRELIMINARY

The training set is $\mathcal{D}_{train} = \{(x_i, y_i, d_i) | \ i \ \in \{1, \ldots, I\}\}$ and the testing set is $\mathcal{D}_{test} = \{x_j | j \in \{1, \ldots, J\}\}$. For generalizable sleep staging, the training data $\mathcal{D}_{train}$ and testing data $\mathcal{D}_{test}$ are from different subjects. $x_i = \left[x_i^e, \ x_i^o, \ x_i^m, \ x_i^c\right]$ and $x_j = \left[x_j^e, \ x_j^o, x_j^m, \ x_j^c\right]$ are PSG signals containing synchronized EEG, EOG, EMG, and ECG, and each epoch lasts 30 seconds. $x_i$ and $x_j$ are the $i$-th and $j$-th samples of training and testing set, respectively. $y_i \in \{1, 2, \ldots, K^y\}$ is the corresponding sleep category and $K^y$ is the total number of categories. $d_i \in \{1, 2, \ldots, K^d - 1\}$ is the domain of $x_i$ and $K^d - 1$ is the number of domain in training set. $I$ and $J$ are the sample sizes of the training and testing set, respectively. $C^U = C^e + C^o + C^m + C^c$ is the total number of PSG channels (containing EEG, EOG, EMG and ECG). $U$ is the number of modalities, $C^e, C^o, C^m$ and $C^c$ are the channel numbers of different modalities.

Sleep staging problem is defined as $\hat{y}_i = \underset{k^y}{\operatorname{argmax}} \left[G_y(G_f(x_i))\right]_{k^y}$, where $G_f = [G_f^e, G_f^o, G_f^m, G_f^c]$ denote modality-specific feature extractors for EEG, EOG, EMG, ECG, respectively, and the parameter of $G_f$ denoted as $\theta^f = \left[\theta^{f,e}, \theta^{f,o}, \theta^{f,m}, \theta^{f,c}\right]$, $G_y$ denotes the label classifier. $\hat{y}_i$ denotes the predicted category and $k^y$ denotes the index of category. $f_i = \left[f_i^e, \ f_i^o, \ f_i^m, \ f_i^c\right] = G_f(x_i) = \left[G_f^e(x_i^e), \ G_f^o(x_i^o), \ G_f^m(x_i^m), \ G_f^c(x_i^c)\right]$ denotes the multimodal fusion feature. $f_i^e, \ f_i^o, \ f_i^m$ and $f_i^c$ represent the features extracted from $x_i^e, \ x_i^o, \ x_i^m$ and $x_i^c$, respectively. We combine the multimodal feature learning model with the domain adversarial method to create the naive multimodal generalizable sleep staging. The predicted domain in domain discrimination is $\hat{d}_i = \underset{k^d}{\operatorname{argmax}} \left[G_d(G_f(x_i))\right]_{k^d}$,

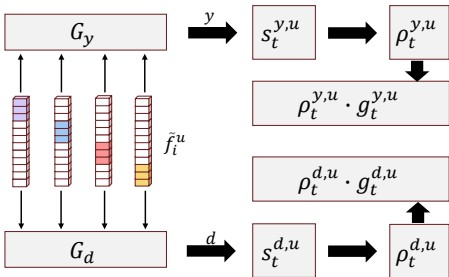

**Figure 3: To calculate the classification and domain discrimination performance metrics for each modality, we employ zero-padding to align the features of each modality with the multimodal fusion feature and feed them to the globally shared classifier and domain discriminator.**

where $G_d$ denotes the domain discriminator and $k^d$ denotes the index of domain.

To simplify the symbol expression, we use $a \in \{y, d\}$ to represent the symbol related to the category and domain. Specially, $K^a \in \{K^y, K^d\}$ denote the number of category and domain, $k^a \in \{k^y, k^d\}$ denote the label index of category and domain, $p_i^a \in \{y_i, d_i\}$ denote the truly label of category and domain, $\hat{p}_i^a \in \{\hat{y}_i, \hat{d}_i\}$ denote the predicted label of category and domain, $G_a \in \{G_y, G_d\}$ denote the classifier and domain discriminator and $\theta^a \in \{\theta^y, \theta^d\}$ denote their model parameters. Since the model structure of $G_a$ is a simple fully connected layer, $\theta^a$ can be split into modality-specific $\theta^{a,u}$. In addition, $u \in \{e, o, m, c\}$ denotes the modality-related symbol. $\hat{p}_i^{a,u} \in \{\hat{p}_i^{a,e}, \hat{p}_i^{a,o}, \hat{p}_i^{a,m}, \hat{p}_i^{a,c}\}$ denote the predicted category and domain labels of different modalities from the same $x_i$. $f_i^u$ denotes the modality-specific feature. $\theta^{f,u}$ denotes the parameter of the modality-specific feature extractor $G_f^u$ and $\theta^{a,u}$ denotes the parameter of the modality-specific part of $G_a$.

## 4 THE PROPOSED SLEEPMG

The pipeline of SleepMG is shown in Figure 2. We introduce SleepMG from three aspects: 1) Naive multimodal generalizable sleep staging (NaiveMG), the backbone of SleepMG. 2) Modal performance metrics of classification and domain discrimination. 3) Metric-guided adaptive modality balancing and generalizable sleep staging.

### 4.1 Naive Multimodal Generalizable Sleep Staging Method

To realize SleepMG, we first conduct the naive integration of modality-specific feature learning and domain adversarial methods, named the Naive Multimodal Generalizable method (NaiveMG). Specifically, NaiveMG employs FeatureNet [18], a dual-scale CNN model to establish each modality-specific feature extractor. These feature extractors capture the unique characteristics of each modality. In addition, NaiveMG fuses extracted features and classifies sleep stages using the simple fully connected classifier $G_y$. To further enhance cross-domain generalization, we set domain discriminator $G_d$ with the same simple fully connected to confuse the identification of the model about domains by back-propagating negative gradients.

By incorporating the above techniques, NaiveMG significantly enhances the cross-domain generalization capability of multimodal sleep staging models, enabling more accurate and reliable sleep stage classification across domains. The classification loss $\mathcal{L}_y$ and domain discrimination loss $\mathcal{L}_d$ are both calculated using the cross-entropy loss function. Details are as follows:

$$\mathcal{L}_a = -\frac{1}{I} \sum_{i=1}^{I} \sum_{k^a=1}^{K^a} 1_{p_i^a=k^a} \log([G_a(f_i)]_{k^a}) \quad (1)$$

where $G_a(f_i) = [G_a(f_i)[1], G_a(f_i)[2], \ldots, G_a(f_i)[K^a]]$ represents the list of softmax predicted output, $i \in \{1, 2, \ldots, I\}$ represents the total number of training samples, 1 represents an indicator function that returns 1 if the inputs are equal and returns 0 otherwise.

All model parameters are optimized by minimizing $\mathcal{L}$. Since $\mathcal{L}_d$ undergoes a gradient reversal in gradient backpropagation, the final optimization process is to minimize $\mathcal{L}_y$ and maximize $\mathcal{L}_d$. The sum of loss $\mathcal{L}$ is as follows:

$$\mathcal{L} = \mathcal{L}_y - \mathcal{L}_d \quad (2)$$

Utilizing NaiveMG as the backbone, SleepMG further quantifies inter-modal differences in classification and domain discrimination.

### 4.2 Multimodal Performance Metric

Figure 3 depicts the details of assessing the modal performance metrics of the classification and domain discrimination.

*4.2.1 Evaluation of Multiple Modalities with Shared Model.* To begin, we employ zero-padding on the modality-specific feature $f_i^u$ to match the desired fused length, denoted as $\tilde{f}_i^u$. Subsequently, these padded features $\tilde{f}_i^u$ are fed into the globally shared classifier $G_y$ and domain discriminator $G_d$, respectively, to obtain their modality-specific prediction results $\hat{p}_i^{a,u}$:

$$\hat{p}_i^{a,u} = \underset{k^a}{\arg\max} \left[ G_a(\tilde{f}_i^u) \right]_{k^a} \quad (3)$$

where $G_a(\tilde{f}_i^u) = \left[ G_a(\tilde{f}_i^u)[1], G_a(\tilde{f}_i^u)[2], \ldots, G_a(\tilde{f}_i^u)[K^a] \right]$ represents the list of softmax predicted output of the modality $u$, $\tilde{f}_i^u$ represents the feature of single modality $u$ that has been zero-padded to match the length of multimodal fusion feature $f_i$.

*4.2.2 Two Performance Values of Each Modality.* Following [35], we calculate the sum of correctly predicted category probabilities from the softmax output as two performance values at the $t$ time step of each modality. The classification and domain discrimination performance values of each modality are as follows:

$$v_t^{a,u} = \frac{1}{I} \sum_{i=1}^{I} \sum_{k^a=1}^{K^a} 1_{p_i^{a,u}=k^a} \left[ G_a(\tilde{f}_i^u) \right]_{k^a} \quad (4)$$

where $t$ is the time step, and 1 is the equality function that returns 1 if the inputs are equal and returns 0 otherwise.

*4.2.3 Two Performance Metrics of Each Modality.* According to two sets of performance values $v_t^{a,u} \in \{v_t^{y,u}, v_t^{d,u}\}$, we calculate the performance score $s_t^{a,u} \in \{s_t^{y,u}, s_t^{d,u}\}$ of each modality at $t$ moment and finally define the modal performance metrics $\rho_t^{a,u} \in \{\rho_t^{y,u}, \rho_t^{d,u}\}$ based on the variance of the difference between $s_t^{a,u}$ and

the average ratio $\frac{1}{U}$ of each modality. The smaller $\rho_t^{a,u}$ means the more balanced modality in classification or domain discrimination:

$$s_t^{a,u} = \frac{v_t^{a,u}}{\sum_{u=1}^{U} v_t^{a,u}} \tag{5}$$

$$\rho_t^{a,u} = tanh\left(\frac{(s_t^{a,u} - \frac{1}{U})^2}{\sum_{u=1}^{U} (s_t^{a,u} - \frac{1}{U})^2}\right) \tag{6}$$

where $tanh$ is the active function for smooth, and the sum scaling method is employed for normalization.

## 4.3 Metric-Guided Modality-Balanced Model Construction

We backpropagate the gradients $g_t^{f,u}$ of the feature extractors normally, computed as $\frac{\partial \mathcal{L}_y}{\partial \theta_t^{f,u}} - \frac{\partial \mathcal{L}_d}{\partial \theta_t^{f,u}}$. While the gradients $g_t^{a,u}$, which are the modality-specific parts from $\frac{\partial \mathcal{L}_a}{\partial \theta_t^a}$, are guided by scalar $\rho_t^{a,u}$ to balance the classification and domain discrimination across modalities:

$$\tilde{g}_t^{a,u} = \rho_t^{a,u} \cdot g_t^{a,u} \tag{7}$$

We utilized the Adam optimizer for all models. Initially, we set the first-moment estimation $m_t^{a,u}$ and the second-moment estimation $v_t^{a,u}$ of the gradient to zero at time zero. Subsequently, we update $m_t^{a,u}$ and $v_t^{a,u}$ at time $t$ based on estimation values of the previous time and the gradient $g_t^{a,u}$ of modality-specific parts of $G_a$:

$$m_t^{a,u} = \beta_1 \cdot m_{t-1}^{a,u} + (1 - \beta_1) \cdot \tilde{g}_t^{a,u} \tag{8}$$

$$v_t^{a,u} = \beta_2 \cdot v_{t-1}^{a,u} + (1 - \beta_2) \cdot (\tilde{g}_t^{a,u})^2 \tag{9}$$

where $g_t^{a,u}$ denote the gradient of modality-specific parts of $\theta_t^a$. $m_t^{a,u}$, $v_t^{a,u}$ denote the first-moment and second-moment estimations of $\tilde{g}_t^{a,u}$, respectively. $\beta_1$ and $\beta_2$ denote the attenuation factors. $(\tilde{g}_t^{a,u})^2$ denotes the square of the gradient $\tilde{g}_t^{a,u}$.

Then further revise the first-moment and second-moment estimations to prevent them from being initially biased toward zero:

$$\hat{m}_t^{a,u} = \frac{m_t^{a,u}}{1 - \beta_1} \tag{10}$$

$$\hat{v}_t^{a,u} = \frac{v_t^{a,u}}{1 - \beta_2} \tag{11}$$

where $\hat{m}_t^{a,u}$ and $\hat{v}_t^{a,u}$ are revised $m_t^{a,u}$ and $v_t^{a,u}$.

The parameters $\theta_t^{a,u}$ of modality-specific parts of $G_a$ are updated according to modal performance metrics $\rho_t^{a,u}$:

$$\theta_{t+1}^{a,u} = \theta_t^{a,u} - \left(\frac{\hat{m}_t^{a,u}}{\sqrt{\hat{v}_t^{a,u}} + \epsilon_1}\right) \cdot \eta \tag{12}$$

where $t$ represents the time step, $\eta$ represents the learning rate, $\epsilon_1$ represents a small constant to prevent division by zero.

By setting a larger $\rho_t^{a,u}$ for the gradient $g_t^{a,u}$ of models $\theta_t^{a,u}$ with poorly-balanced modality in classification and domain discrimination performance and a smaller $\rho_t^{a,u}$ for the gradient $g_t^{a,u}$ with well-balanced modality, we achieve a multimodal generalizable sleep staging model that is modality-balanced. Notably, Since the gradients backpropagated through the domain discriminator undergo the same reversal, the balance degree of modalities remains unaffected.

## 4.4 Method Implement

The core of SleepMG is elucidated in Algorithm 1. SleepMG aims to calculate the performance of multiple modalities' classification and domain discrimination, ensuring the inter-modal balance in both classification and domain discrimination and achieving multimodal generalizable sleep staging. The backbone is naiveMG, which includes multimodal feature learning and domain adversarial methods. Multimodal PSG is first processed through modality-specific feature extractors $G_f^u$ to extract features and fuse. Subsequently, domain adversarial methods are integrated to improve cross-domain generalization. The model is optimized by minimizing classification loss $\mathcal{L}_y$ and maximizing domain discrimination loss $\mathcal{L}_d$. By the early convergence stage, we assess the classification and domain discrimination capabilities of each modality by zero-padding their features and utilizing the shared classifier and discriminator. Additionally, the parameters $\theta_t^{f,u}$ of $G_f^u$ are updated with minimizing $\mathcal{L}$ as usual. Furthermore, we calculate modal performance metrics of classification and domain discrimination, leveraging these metrics to influence the gradient update of parameters $\theta_t^a$ of $G_a$ with minimizing $\mathcal{L}$. To achieve modality balance, the gradients backpropagated by modality-specific parts $\theta_t^{a,u}$ that correspond to the poorly-balanced modalities are relatively enhanced with large $\rho_t^{a,u}$, while those that correspond to the well-balanced modalities are relatively reduced with small $\rho_t^{a,u}$. The balanced multimodal generalizable sleep staging model is obtained when all the models converge.

---

**Algorithm 1** The whole process of SleepMG method

---

**Input:** The training set $\mathcal{D}_{train} = \{(x_i, y_i, d_i) | i \in \{1, \dots, I\}\}$, the testing set $\mathcal{D}_{test} = \{x_j | j \in \{1, \dots, J\}\}$.

**Output:** The sleep staging result of testing data $\hat{y}_j$.

1: Initialize model parameters $\theta_{t=0}^{f,u}$ of the modality-specific feature extractor $G_f^u$ and $\theta_{t=0}^{a,u}$ of modality-specific parts of $G_a$(i.e., classifier and domain discriminator);

2: **repeat**

    % **Metric Definition**

3:    Extract the multimodal feature of $x_i$ with $G_f$: $f_i = G_f(x_i)$;

4:    Zero-pad $f_i^u$ to obtain $\tilde{f}_i^u$ and calculate two performance values of each modality $v_t^{a,u}$ with Eq. (4);

5:    Calculate two performance metrics $\rho_t^{a,u}$ of each modality based on $v_t^{a,u}$ with Eq. (5) and (6);

    % **Model Training**

6:    Calculate total loss $\mathcal{L}$ with Eq. (2);

7:    Update model parameters $\theta_t^{f,u}$ normally, while update $\theta_t^{a,u}$ under the guidance of $\rho_t^{a,u}$ using Eq. (12) to minimize $\mathcal{L}$;

8: **until** Iterate until model convergence

9: Sleep staging prediction: $y_j = G_y(f_j)$

---

## 5 EXPERIMENTS

All experiments are implemented with Python 3.8.5 and Pytorch 1.7.1. We conduct them on a computer server with 640GB RAM and two NVIDIA RTX A5000 GPUs with 24GB VRAM each.

## 5.1 Dataset and Data Processing

We evaluate SleepMG on two public datasets: ISRUC-S3 [23], and MASS-SS3 [31]. For a fair comparison, we remove the Movement and Unknown stages and merge the N3 and N4 stages into a single N3 stage, resulting in five categories of Wake, N1, N2, N3, and REM sleep stages according to the AASM standard. Each PSG recording is downsampled to 100 Hz and divided into 30-second epochs.

**ISRUC-S3** collects PSG from 10 subjects (one male and nine female) over 8 hours in a single night, containing 8589 sleep epochs. We select ten of the 12 channels and remove the leg EMG far from the brain. It comprises four modalities: six-channel EEG, two-channel EOG, one-channel chin EMG, and ECG.

**MASS-SS3** collects PSG from 62 subjects (28 male and 34 female) over 8 hours in a single night, containing 59304 sleep epochs. We select the same 10 channels with four modalities as ISRUC-S3.

## 5.2 Experiment Settings and Implementation

In the experiment, we utilize FeatureNet [16, 18] as the backbone of NaiveMG and SleepMG. We employ the leave-one-subject-out method to divide the dataset into five domains with different subjects and perform five-fold cross-validation to split the data into training and testing sets. From the training set, we further allocate 20% as a validation set. The best model based on the validation set is saved and evaluated on the cross-domain testing set. The final reported results represent the overall evaluation of the entire dataset. We employ the Adam optimizer for model optimization with a learning rate of 0.001, attenuation factors $\beta_1$ of 0.5 and $\beta_2$ of 0.999. The batch size for data processing is 256, and the training process is conducted over 50 epochs. The gradient reversal ratio employed in domain-adversarial training for the ISRUC-S3 and MASS-SS3 datasets is 0.1 and 1, respectively.

We employ *Accuracy*, *Macro F*1, *Kappa*, and *F*1 for each category as the evaluation metrics of the experimental results. *Accuracy* measures the proportion of correctly classified sleep epochs. The *F*1 score for each category evaluates precision and recall, comprehensively evaluating the performance of the model. *Macro F*1 score calculates the average F1 score across all sleep stage categories, providing insight into the capability of the model to perform well across categories without favoring a specific category. *Kappa* measures the agreement between the prediction of the model and the ground truth, accounting for chance. It assesses inter-rater agreement by considering both observed and expected accuracy.

## 5.3 Baseline Methods and Settings

We select eight classical and state-of-the-art sleep staging methods as baseline methods from four perspectives: four conventional sleep staging [18, 48, 64, 10] method, one generalizable sleep staging [50] method is based on transfer learning to improve cross-domain sleep staging, two multimodal sleep staging [17, 60] methods involve modality-specific feature extractors, and one balance method [35] is special-designed for multimodal fusion and balance.

- FeatureNet [18]: The conventional method uses a dual-scale CNN structure with varying kernel sizes and channel configurations. The first set of kernels consists of sizes {50, 8, 8, 8} and corresponding channels {32, 64, 64, 64}. The second set

of kernels consists of sizes {64, 8, 6, 6, 4} and corresponding channels {64, 64, 64, 64}.
- DeepSleepNet [48]: The conventional method employs the CNN-BiLSTM model. CNNs captures local temporal patterns and spatial relationships, while the BiLSTM component captures long-term temporal dependencies.
- MaskSleepNet [64]: The conventional method combines dual-scale CNNs for feature extraction, squeeze and excitation block for optimizing feature weights, and multi-head attention for capturing temporal information.
- AttnSleep [10]: The conventional method integrates a multi-resolution CNN model, adaptive feature recalibration for learning interdependencies among features, and multi-head attention for capturing temporal dependencies.
- DAN [50]: The generalizable method is based on CNN-BiGRU and leverages the MMD alignment to learn domain-invariant representation.
- SleepPrintNet [17]: The multimodal method involves modality-specific feature learning based on the 1D convolutional model. In addition to learning temporal features across all PSG modalities, the EEG modality specifically focuses on spectral-spatial feature learning.
- MMASleepNet [60]: The multimodal method incorporates modality-specific feature learning through 1D convolutional models and squeeze-and-excitation, along with feature fusion utilizing transformer encoder models.
- OGM-GE [35]: The balance method was originally an audio-visual dual-modality balancing method, which improved the model by balancing the classification performance of two modalities and modulating the gradient of the feature extractor. To be fair, we employ FeatureNet as its backbone.

## 5.4 Comparative Experiment Results

To demonstrate the advantage of SleepMG in sleep staging, we compared SleepMG with a total of eight methods, including four conventional methods, one generalizable method, two multimodal methods, and one multimodal balance method. The experimental results in Table 1 show that SleepMG consistently outperforms other methods on both datasets. On the ISRUC-S3 dataset, the second-best *Accuracy* method is the NaiveGS method, DAN, and SleepMG achieves an *Accuracy* improvement of nearly two percentage points compared to it and approximately 3.6% *Accuracy* improvement compared to the backbone method, FeatureNet. These results indicate the effectiveness of transfer learning methods in enhancing cross-domain sleep staging performance, and SleepMG demonstrates superior performance by balancing modalities in classification and domain discrimination. On the MASS-SS3 dataset, the second-best method is the modal balancing method (OGM-GE), and SleepMG further improves by approximately 1.0% compared to it, highlighting the importance of modality balancing and the greater role of SleepMG in balancing the two performances of modalities.

## 5.5 Imbalance Degrees Visualization Results

As illustrated in Figure 4, to further validate our progress in achieving modality balance, we visualized the changes of modal imbalance degree during training on the ISRUC-S3 dataset. The modal

Table 1: The performance comparison of state-of-the-art methods and SleepMG on two public datasets. The bold and underline items denote the best and second-best results, respectively.

| Dataset | Method | Overall results | | | F1 for each category | | | | |
|---|---|---|---|---|---|---|---|---|---|
| | | Accuracy | Macro F1 | Kappa | Wake | N1 | N2 | N3 | REM |
| ISRUC-S3 | FeatureNet [18] | 0.7513 | 0.7275 | 0.6803 | 0.8456 | 0.4841 | 0.7409 | 0.8531 | 0.7136 |
| | DeepSleepNet [48] | 0.7456 | 0.7393 | 0.6758 | 0.8782 | 0.5306 | 0.7040 | 0.8277 | 0.7560 |
| | MaskSleepNet [64] | 0.6571 | 0.6297 | 0.5637 | 0.6516 | 0.4751 | 0.6903 | 0.8390 | 0.4925 |
| | AttnSleep [10] | 0.7611 | 0.7404 | 0.6932 | 0.8537 | 0.5041 | 0.7558 | 0.8649 | 0.7234 |
| | DAN[50] | 0.7687 | 0.7442 | 0.7008 | 0.8461 | 0.4484 | **0.7711** | **0.8752** | 0.7802 |
| | SleepPrintNet [17] | 0.7591 | 0.7435 | 0.6889 | 0.8305 | 0.5162 | 0.7565 | 0.8530 | 0.7614 |
| | MMASleepNet [60] | 0.7552 | 0.6847 | 0.6831 | 0.8588 | 0.2041 | 0.7573 | 0.8590 | 0.7442 |
| | OGM-GE [35] | 0.7610 | 0.7472 | 0.6947 | 0.8709 | 0.5415 | 0.7207 | 0.8386 | 0.7644 |
| | **SleepMG** | **0.7868** | **0.7745** | **0.7264** | **0.8833** | **0.5691** | 0.7613 | 0.8641 | **0.7950** |
| MASS-SS3 | FeatureNet [18] | 0.8533 | 0.8019 | 0.7827 | 0.8968 | 0.5120 | 0.8964 | 0.8395 | 0.8649 |
| | DeepSleepNet [48] | 0.8531 | 0.7984 | 0.7807 | 0.8853 | 0.5083 | 0.9017 | 0.8380 | 0.8585 |
| | MaskSleepNet [64] | 0.8296 | 0.7679 | 0.7490 | 0.8573 | 0.4325 | 0.8818 | 0.8138 | 0.8541 |
| | AttnSleep [10] | 0.8510 | 0.7960 | 0.7796 | 0.8918 | 0.4836 | 0.8940 | 0.8488 | 0.8618 |
| | DAN [50] | 0.8231 | 0.7380 | 0.7326 | 0.8432 | 0.3323 | 0.8806 | 0.8150 | 0.8192 |
| | SleepPrintNet [17] | 0.8459 | 0.7871 | 0.7702 | 0.8816 | 0.4674 | 0.8859 | 0.8420 | 0.8584 |
| | MMASleepNet [60] | 0.8405 | 0.7820 | 0.7598 | 0.9001 | 0.5055 | 0.8822 | 0.7390 | 0.8831 |
| | OGM-GE [35] | 0.8570 | 0.8033 | 0.7881 | 0.8906 | 0.5131 | 0.9004 | 0.8407 | 0.8715 |
| | **SleepMG** | **0.8660** | **0.8169** | **0.8015** | **0.9005** | **0.5462** | **0.9037** | **0.8493** | **0.8847** |

imbalance degrees of classification and domain discrimination performances are measured with $\sum_{u=1}^{U}(s_t^{a,u} - \frac{1}{U})^2$, the sum of squared mean relative differences [2, 47]. The lower the modal imbalance degree, the more balanced each modality will be. It can be observed that the SleepMG method exhibits lower modal imbalance in both classification and domain discrimination, indicating a better balance. Additionally, we visualized the changes in test accuracy in Figure 5, revealing a significant improvement in cross-domain sleep staging on the test set when the modalities are more balanced.

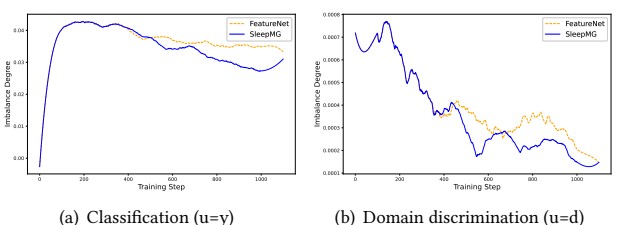

(a) Classification (u=y)          (b) Domain discrimination (u=d)

Figure 4: The change of the modal imbalance degrees of (a) Classification and (b) Domain discrimination during training on the ISRUC-S3 dataset.

## 5.6 Ablation Experiment Results

As is shown in Table 2, to further demonstrate the contribution of each module in SleepMG, we compared the effects of multimodal and generalizable modules of NaiveMG (✓✓ in Table 2). We employ FeatureNet [18] as the backbone (✗✗ in Table 2).The multimodal module performs modality-specific feature extraction followed by direct concatenation fusion. The generalizable module enhances model generalization by employing the domain adversarial learning method. The experimental results of sleep staging on two datasets

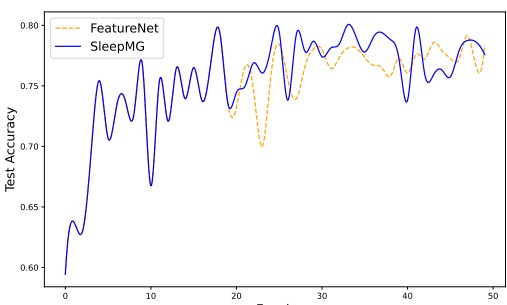

Figure 5: The change of the test accuracy per epoch on the ISRUC-S3 dataset with FeatureNet and SleepMG.

show that the generalization method provides a little improvement, while the multimodal module exhibits a more noticeable effect compared to the generalizable module alone. Furthermore, the combination of multimodal and generalizable modules leads to a significant improvement.

To explore effective feature fusion methods for multimodal sleep staging, we employ FeatureNet to extract single-modal features and fuse them with direct concatenation, self-attention weighting, and modality balancing approaches, respectively. From the experimental results in Table 3, it can be seen that self-attention performed worse than direct concatenation. Conversely, the modality balancing method performed well on both datasets. This suggests self-attention focuses more on well-performing modalities, exacerbating modality imbalances and harming classification. Furthermore, contrary to the attention-based approach, the balancing method assigns greater weight to poorly performing modalities, positively contributing to classification accuracy.

**Table 2: Ablation study of M(ultimodal) and G(eneralizable) modules of NaiveMG(✓✓) on two public datasets. The bold and underline items denote the best and second-best results, respectively.**

| Dataset | M | G | Overall results | | |
|---|---|---|---|---|---|
| | | | Accuracy | Macro F1 | Kappa |
| ISRUC-S3 | × | × | 0.7513 | 0.7275 | 0.6803 |
| | × | ✓ | 0.7545 | 0.7323 | 0.6844 |
| | ✓ | × | 0.7610 | 0.7487 | 0.6927 |
| | ✓ | ✓ | **0.7711** | **0.7592** | **0.7067** |
| MASS-SS3 | × | × | 0.8533 | 0.8019 | 0.7827 |
| | × | ✓ | 0.8544 | 0.8017 | 0.7841 |
| | ✓ | × | 0.8570 | 0.8058 | 0.7869 |
| | ✓ | ✓ | **0.8571** | **0.8068** | **0.7876** |

**Table 3: Ablation study of FeatureNet with different multimodal feature fusion methods on two public datasets. The bold and underline items denote the best and second-best results, respectively.**

| Dataset | Concat Method | Overall results | | |
|---|---|---|---|---|
| | | Accuracy | Macro F1 | Kappa |
| ISRUC-S3 | Direct-concat | 0.7610 | 0.7487 | 0.6927 |
| | Self-attention | 0.7550 | 0.7490 | 0.6885 |
| | **Balance** | **0.7738** | **0.7587** | **0.7089** |
| MASS-SS3 | Direct-concat | 0.8570 | 0.8058 | 0.7869 |
| | Self-attention | 0.8558 | 0.8060 | 0.7858 |
| | **Balance** | **0.8603** | **0.8121** | **0.7936** |

**Table 4: Ablation study of NaiveMG(× × ×) with balancing different components on public datasets. The bold and underline items denote the best and second-best results, respectively.**

| Dataset | $G_f^u$ | $G_y^u$ | $G_d^u$ | Overall results | | |
|---|---|---|---|---|---|---|
| | | | | Accuracy | Macro F1 | Kappa |
| ISRUC-S3 | × | × | × | 0.7711 | 0.7592 | 0.7067 |
| | ✓ | × | × | 0.7680 | 0.7590 | 0.7034 |
| | × | ✓ | × | 0.7793 | 0.7647 | 0.7168 |
| | × | × | ✓ | 0.7728 | 0.7600 | 0.7085 |
| | **×** | **✓** | **✓** | **0.7868** | **0.7745** | **0.7264** |
| MASS-SS3 | × | × | × | 0.8571 | 0.8068 | 0.7876 |
| | ✓ | × | × | 0.8582 | 0.8071 | 0.7890 |
| | × | ✓ | × | 0.8620 | 0.8117 | 0.7951 |
| | × | × | ✓ | 0.8634 | 0.8126 | 0.7975 |
| | **×** | **✓** | **✓** | **0.8660** | **0.8169** | **0.8015** |

As shown in Table 4, we compared the effects of balancing different components of NaiveMG (×××  in Table 4). The results indicate that balancing the modality-specific feature extractor has minimal or no effect, while the fine-grained inter-modal balance of the

modality-specific parts of classifier $G_y$ and domain discriminator $G_d$ has positive effects. In multimodal feature learning, focusing solely on achieving modality balance may significantly weaken the model's ability to extract modality-related information. In domain adversarial learning, classification capability and domain discrimination capability are often seen as opposing objectives. However, both imbalances across modalities can negatively affect the performance of the model. For example, if there is a significant difference in domain discrimination capability between different modalities, the model may learn irrelevant information about the participants that is modality-dependent or modality-specific, which can limit the capability of the model to generalize across domains. This inter-modal imbalance can be detrimental to performance. In summary, these two inter-modal imbalances can negatively affect model performance. The fine-grained inter-modal balance of the classification and domain discrimination capabilities is crucial to mitigate these negative effects and improve overall performance in sleep staging.

## 5.7 Feature Visualization Analysis

For more interpretable analysis, we exploit the t-SNE [55] to visualize the feature embeddings of Feature and SleepMG methods. As shown in Figure 6, SleepMG exhibits more distinct classification boundaries than the FeatureNet method. For example, in SleepMG, the cluster representing the deep sleep stage N3 is fewer in number and more tightly grouped, while the cluster representing the N2 stage (indicated by green) clearly separates the blue N3 stage from other stages. The light sleep stage N1, which is the most challenging to differentiate, exhibits a tightly grouped yellow cluster in SleepMG, displaying clearer boundaries.

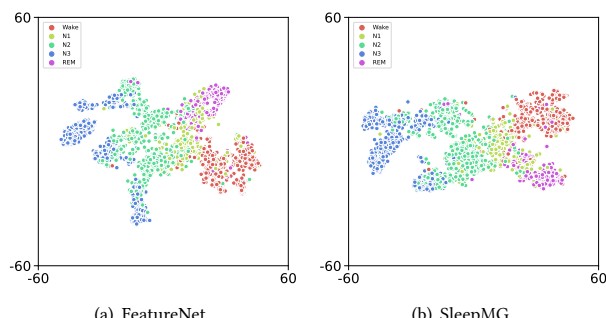

(a) FeatureNet          (b) SleepMG

**Figure 6: The visualization of the t-SNE embeddings on the ISRUC-S3 dataset with (a) FeatureNet and (b) SleepMG.**

## 6 CONCLUSION

This paper introduces SleepMG, a novel Multimodal Generalizable Sleep Staging method. Besides the naive integration of multimodal feature learning models and domain generalization methods, SleepMG quantitatively assesses and adaptively balances the classification and domain discrimination capabilities of multiple modalities, addressing the issues of inter-modal imbalances. Experimental results demonstrate that SleepMG outperforms state-of-the-art performance in cross-domain sleep staging and achieves inter-modal balances.

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
