# OpenReview forum: "SleepMG: Multimodal Generalizable Sleep Staging with Inter-modal Balance of Classification and Domain Discrimination"
_acmmm.org/ACMMM/2024/Conference — MM2024 Oral_

### Official Review · Reviewer_wwA5 · 2024-05-16

**Rating:** 6
**Confidence:** 4

**Summary:**

This paper introduces a novel method for sleep staging called SleepMG, which is designed to balance inter-modal differences and achieve highly accurate cross-domain sleep staging.  Experimental results show that SleepMG outperforms other eight state-of-the-art sleep staging methods, effectively balancing multiple modalities across two PSG datasets.

**Strengths:**

1. This paper is well constructed and written, with a high level of clarity. The ablation study is sufficient and fully convincing.
2. SleepMG leverages multiple modalities (EEG, EOG, EMG, ECG) to improve the accuracy of sleep staging, providing a more comprehensive approach compared to using a single modality.
3. The mathematical modeling and algorithmic development are both sound and robust. The use of adaptive gradients based on performance metrics is particularly noteworthy for its potential to optimize learning dynamically.
4. SleepMG enhances cross-domain generalization, which is crucial for applying the model to new, unseen subjects.
5. The development of "SleepMG," which assesses and adjusts for modal imbalances in real-time, is a significant contribution. This approach not only advances the state-of-the-art in sleep staging but also offers a potential framework for other multimodal challenges.

**Limitations:**

1. Displaying the confusion matrix would be useful for further explaining the results.
2. Figure 1 does not seem particularly useful.
3. Highlighting future work on multi-modal and inter-modal balance is favorable.

**Suitability:**

3

---

### Official Review · Reviewer_6XUb · 2024-05-20

**Rating:** 5
**Confidence:** 4

**Summary:**

This paper proposes SleepMG, which introduces two features into the modeling of PSG signals, (1) inter-model balance and (2) domain generalization. Extensive experiments validate their models' effectiveness and the variants of SleepMG.

**Strengths:**

1. The motivation of this paper is well-defined and sufficiently supported.
2. The method of introducing an imbalance score into the modeling of different modalities is novel.
3. The experiments and analysis are sufficient.
4. The presentation of this paper is good.

**Limitations:**

1. The author proposes a cross-modality method but only tests in one setting: i.e., six-channel EEG, two-channel EOG. one-channel chin EMG, and ECG. More experimental setups of combinations of modalities should be considered. For example, using all the channels in MASS-SS3 (I think it is not necessary to select the same 10 channels in MASS-SS3) or using different datasets such as SHHS and CFS which contain more data samples.

2. The author adopts a domain classifier to force the model to learn domain-invariant representation for sleep staging. The design is reasonable but only if the number of participants is small. If the number of subjects is large, e.g., 1737 subjects in the SHHS dataset, and 320 subjects in the CFS dataset, it will be very difficult for the classifier to make good predictions. On the other hand, the model has to change its structure when the number of subjects increases in a continual training setting. One solution is using a loss function to guide the model to distinguish from different modalities (e.g., contrastive learning).

3. Some small issues regarding the paper presentation, e.g., the text in Figure 4 is too small, lacks a notation table in the method section (there are too many notations in this paper), and Figure 2 is difficult to understand before we carefully read the method section.

**Suitability:**

3

---

### Official Review · Reviewer_Rg5K · 2024-05-22

**Rating:** 3
**Confidence:** 3

**Summary:**

This paper presents SleepMG that detects the sleep stage of a subject using multimodal signals (i.e., EEG, EOG, EMG and ECG). Especially, SleepMG focuses on balancing the modalities in terms of their classification and domain discrimination performances. For this purpose, the modal performance metrics are devised and used to weight gradients of the modality-specific parts. The experimental results on two public datasets ISRUC-S3 and MASS-SS3 show the effectiveness of SleepMG.

**Strengths:**

- The idea to balancing modalities seems interesting.
- Especially, balancing modalities is considered on not only the classification performance of each modality but also the discrimination performance, in order to strengthen the domain-invariant power (cross-subject generality capability) of SleepMG.
- Intensive experiments have been conducted to show the effectiveness of SleepMG.
- This paper is well-written.
- Although it is not a contribution of this paper, domain adversarial learning looks useful for extracting domain-invariance (i.e., subject-invariance) features.

**Limitations:**

- I'm not completely convinced with the importance of balancing modalities. In particular, I'm wondering if it may weaken the effect of a very useful modality so that the final sleep staging performance may be degraded. Is it possible to theoretically proof that balancing modalities is better than biasing certain modalities?
- I'm not sure if the proposed modal performance metrics really lead to balancing modalities. This is because $\rho_t^{a,u}$ by Eq. (6) $ seems to be a normalised $v_t^{a,u}$ in Eq. (4) representing the sum of correct predictions of modality $u$. I'm wondering why using such $\rho_t^{a,u}$ to weight gradients is useful for balancing modalities.
- There seems no explanation about the "well-balanced" or "poorly-balanced" situations. In addition, how is the "well-balanced" situation different from "well-performing" situtation?
- In Fig. 5, the final performance of FeatureNet seems to be better than that of SleepMG. Does this figure really presents the effectiveness of balancing modalities? Also, considering Fig. 4, I'm wondering whether or not the modality imbalance is really a problem after a sufficient number of training epochs.

**Suitability:**

3

---

### Official Review · Reviewer_1PkZ · 2024-05-26

**Rating:** 2
**Confidence:** 3

**Summary:**

This paper presents a multi-modal based sleep staging method, SleepMG. It aims to address the inter-modal balance by defining the performance metrics of individual modalities and utilizing such metrics to adaptively adjust the gradients of classifiers and domain discriminator. Experiments were conducted on two public datasets: ISRUC-S3 and MASS-SS3.

**Strengths:**

1) The proposed method is reasonable with a clear motivation
2) Experimental results are comprehensive with rich comparisons, good ablation studies, and various discussions.

**Limitations:**

1) While the research motivation is clear, the proposed solution lacks clear contributions, as the gradient adjustment looks quite empirical. I view this study as a multi-modal fusion task and would like to see strong argument along this point.
2) While the authors argue for generalization, there is no clear emphasis at the end of Introduction for summarizing contributions.
3) Comparisons with other (larger) datasets such as SleepEDF with nearly 80 subjects are needed. In addition, the current improvement seems marginal (e.g., less than 0.01).
4) I feel that Figure 1 is too simple to provide much useful context.
5) Language improvement: “modal differences” may be “modality differences”, “to do the subject personalized calibration”, “none consider(s)”, “gradients of model of the classifier”, “modality-balanced”, more explanations on Figure 2, …

**Suitability:**

2

---

### Meta-Review · Area_Chair_da2G · 2024-07-01

**Recommendation:** Accept (Oral)
**Confidence:** 5

**Metareview:**

The paper introduces a novel method for sleep staging called SleepMG, which is designed to balance inter-modal differences and achieve highly accurate cross-domain sleep staging.

After considering the paper, the reviewer's comments, and the rebuttal I recommend 'accept' for the paper.

The reviewers highlight the following strengths and limitations:

Strengths:
1.  Experimental results show that SleepMG outperforms other eight state-of-the-art sleep staging methods, effectively balancing multiple modalities across two PSG datasets.
2. Intensive experiments have been conducted to show the effectiveness of SleepMG.
3. This paper is well-organized and written.

Limitations:
1. One reviewer considers the proposed method impractical
2. Some reviewers find particularly Figure 1 does not seem useful.